# Evaluation of a Proportional–Integral–Derivative Controller for Hemorrhage Resuscitation Using a Hardware-in-Loop Test Platform

**DOI:** 10.3390/jpm12060979

**Published:** 2022-06-16

**Authors:** Eric J. Snider, David Berard, Saul J. Vega, Guy Avital, Emily N. Boice

**Affiliations:** 1U.S. Army Institute of Surgical Research, JBSA Fort Sam Houston, San Antonio, TX 78234, USA; david.m.berard3.ctr@mail.mil (D.B.); saul.j.vega.ctr@mail.mil (S.J.V.); guy.avital.md.il@gmail.com (G.A.); emily.n.boice.ctr@mail.mil (E.N.B.); 2Trauma & Combat Medicine Branch, Surgeon General’s Headquarters, Israel Defense Forces, Ramat-Gan 5262000, Israel; 3Division of Anesthesia, Intensive Care & Pain Management, Tel-Aviv Sourasky Medical Center, Tel-Aviv 6423906, Israel

**Keywords:** control systems, hemorrhage shock, fluid resuscitation, closed-loop, infusion, controllers, hemorrhage resuscitation

## Abstract

Hemorrhage is a leading cause of preventable death in trauma, which can often be avoided with proper fluid resuscitation. Fluid administration can be cognitive-demanding for medical personnel as the rates and volumes must be personalized to the trauma due to variations in injury severity and overall fluid responsiveness. Thus, automated fluid administration systems are ideal to simplify hemorrhagic shock resuscitation if properly designed for a wide range of hemorrhage scenarios. Here, we highlight the development of a proportional–integral–derivative (PID) controller using a hardware-in-loop test platform. The controller relies only on an input data stream of arterial pressure and a target pressure; the PID controller then outputs infusion rates to stabilize the subject. To evaluate PID controller performance with more than 10 controller metrics, the hardware-in-loop platform allowed for 11 different trauma-relevant hemorrhage scenarios for the controller to resuscitate against. Overall, the two controller configurations performed uniquely for the scenarios, with one reaching the target quicker but often overshooting, while the other rarely overshot the target but failed to reach the target during severe hemorrhage. In conclusion, PID controllers have the potential to simplify hemorrhage resuscitation if properly designed and evaluated, which can be accomplished with the test platform shown here.

## 1. Introduction

Hemorrhage remains a leading preventable cause of death for trauma [1], especially in combat casualty care situations [2]. Loss of blood volume causes a decrease in cardiac output (CO), which impairs delivery of oxygen to tissue (DO_2_), leading to the accruement of an oxygen debt that is associated with a poor outcome [3]. It can be minimized with timely restoration of blood volume, using crystalloids or, preferably, blood products or whole blood [4]. Before appropriate hemorrhage control is assured, this should usually be balanced against the need to maintain a certain level of permissive hypotension, to decrease the risk of re-bleeding [5]. As a result, hemorrhagic shock resuscitation is a tedious, high-cognitive-burden task requiring constant adjustments with high inter- and intra-subject variability to fluid delivery rates to maintain patients in close proximity to the therapeutic goal, which is only made more challenging in high-stress, resource-limited trauma situations.

Automated fluid administration systems have the potential to greatly simplify medical care for hemorrhage resuscitation. Closed-loop systems rely on single or multiple input signals, where a controller compares the input signal to desired set points, and output adjustments are made to reach the set point. These adjustments can be provided as decision support [6], as provider-in-loop, or via a fully automated format. While control systems are widely used in everyday life and are becoming more common in medical systems such as insulin delivery [7], ventilation [8], burn fluid resuscitation [9] and sedation [10], systems are also being developed for hemorrhage resuscitation relying on simple table lookup methodologies [11], proportional–integral–derivative (PID) controllers [12], or more advanced adaptive controllers tracking physiological responsiveness [13,14]. However, their development use case is primarily for managing fluid administration in a more controlled surgical setting, as opposed to trauma scenarios where the hemorrhage extent and injury severity are typically unknown, and the therapeutic goals are often different.

Thus, fluid administration controllers for trauma scenarios must be designed to adapt and personalize to much different situations than those needed for surgical systems. These scenarios include varying time spans from injury to treatment initiation, active hemorrhages of different rates, different infusate types and various levels of coagulation system function. Towards this, we have developed a hardware-in-loop automated testbed for resuscitation controllers (HATRC) [15] that allows for setting a range of hemorrhage situations in a physical test platform. HATRC’s volume responsiveness is set by a custom-designed hydrostatic reservoir, termed the PhysioVessel [16], and hemorrhage rate, coagulation profile and initial blood pressure can all be tailored to special scenarios, allowing individual controller evaluation in a range of circumstances and patient variabilities.

Here, we evaluate a closed-loop controller for its ability to administer fluids and stabilize the HATRC platform at a target pressure across eleven testing scenarios. A PID controller was selected for this work as they have been previously used for hemorrhage resuscitation [12,17] and are widely deployed for other control system applications [18,19,20]. The PID controller was tuned for two resuscitation rates (referred to as “conservative” and “aggressive”) to the swine hemorrhage data sets which HATRC mimics, followed by comparison using a number of conventional control system performance metrics.

## 2. Materials and Methods

### 2.1. PID Controller Design and Tuning

Hemorrhage data was compiled from a previous swine study [16] and analyzed using MATLAB (MathWorks, Natick, MA, USA) to assess the relationship between bolus infusion volume of whole blood (WB) and mean arterial pressure (MAP). Animal subjects underwent a controlled hemorrhage of 24 mL/kg followed by a spleen injury prior to the bolus infusion. The system identification toolbox in MATLAB was used to create a plant model for tuning two PID controller configurations. From the swine study data, a representative sample was evaluated during the period between the introduction of the spleen injury and the completion of the infusion. A time-domain data object was created using the infusion rate as the input and calculated MAP as the output. Although a rapid drop in MAP was observed in all datasets immediately following the cessation of the infusion, this was presumed to be a physiological artifact of rapid infusion effects on mean systemic filling pressure that was not replicated in the test platform. MAP was therefore artificially held constant for a period of five minutes after resuscitation was completed for the plant model, to represent the testing conditions more accurately. Next, a transfer function was estimated using a single pole process model with a first-order auto-regressive moving average (ARMA) disturbance applied. The resulting transfer function was used in conjunction with the PID tuner toolbox to tune two discrete-time PID controller objects. An aggressive configuration was created using a response time of 75 s and transient behavior setting of 0.7, and a conservative configuration used a response time of 150 s and transient behavior of 0.75. The aggressive settings were chosen to reduce the rise time of the aggressive controller without exceeding an estimated overshoot of approximately 7.7% (i.e., if target pressure is 65 mmHg, overshoot will not exceed 70 mmHg) and the conservative configuration, by doubling the rise time with a slight shift in transient behavior, reduced the estimated overshoot to approximately 0.2%.

### 2.2. Hardware-in-Loop Automated Testbed for Resuscitation Controllers

HATRC consisted of a physical testing component for mimicking swine physiological volume responsiveness, coupled with computer code for setting precise hemorrhage testing scenarios [15]. Briefly, the physical component of HATRC incorporated a peristaltic pump for water circulation through a loop at 145 mL/min and a pressure transducer (ICU Medical, San Clemente, CA, USA) connected by a three-way valve for monitoring arterial-mimicking water pressure (Figure 1). The loop included two PhysioVessels that acted as the venous capacitance for the system: one for mimicking whole blood (PV_WB_) and the second for mimicking crystalloid (PV_Crys_). We have previously shown that these PhysioVessels were constructed based on pressure–volume relationship functions derived from experimental swine hemorrhagic shock resuscitation data, such that the geometry of the PhysioVessel causes hydrostatic pressure to mimic volume responsiveness during fluid resuscitation with different fluids [16]. This empiric approach was meant to circumvent the need for a complex physiological model but it has its limitations due to the data it is based on, meaning its use is limited to healthy swine subjects with their volume status in the range between profound shock (loss of over 35% of estimated blood volume) and damage-control resuscitation goals, with the resuscitation performed using either whole blood or crystalloids. PV_WB_ and PV_Crys_ were attached to the circulatory loop via a solenoid valve (Grainger, Lake Forest, IL, USA) so that a computer could select which of the two PhysioVessels was actively connected to HATRC at any given time. For fluid infusion and outflow, two additional peristaltic pumps were used and connected directly to the PhysioVessels. Each pump could be diverted to either PhysioVessel using two additional solenoid valves also under computer control.

HATRC’s computer control capabilities were programmed using MATLAB. A USB interface (U3, LabJackCorp, Lakewood, CO, USA) was used to energize the solenoid valves to control PhysioVessel swapover, and infusion/outflow rates were set by sending control commands to the corresponding peristaltic pump via a standard serial port. Infusion was controlled by a PID controller, as described above, and outflow settings for HATRC were set according to the hemorrhage scenarios detailed below (Section 2.3). For all scenarios, a MAP of 65 mmHg was set as the target for PID controlled infusion, as this is similar to remote damage control resuscitation (RDCR) guidelines for a systolic pressure of 85 mmHg in terms of MAP [21]. A data acquisition system (PowerLab, ADinstruments, Sydney, Australia) was used to capture “arterial” pressure waveform data from HATRC. This was recorded at 40 Hz and averaged over a moving five second window to calculate the MAP throughout.

#### 2.2.1. Overview of HATRC Outflow Logic

Outflow from HATRC was a factor of two major components. The first was urine rate, which was held constant at 1.4 mL/min unless MAP was less than 50 mmHg, at which point the urine rate was zero. The second component was hemorrhage rate, which was a combination of a MAP-dependent hemorrhage factor and a hemostasis factor. The hemorrhage factor was set to achieve zero hemorrhage at 30 mmHg MAP and then linearly increase to as high as 70 (slow bleed) or 140 (fast bleed) mL/min at 65 mmHg. Hemorrhage rates were reduced with time by using a time-dependent hemostasis factor, slowing the rate the longer the bleeding occurred in some scenarios. However, this hemostasis feature was disabled when coagulopathy situations were mimicked. Lastly, an overshoot penalty was added into the HATRC logic where, if the measured MAP exceeded the target MAP by more than 5%, both the hemorrhage and hemostasis factors were reset to their starting values, increasing the re-bleed rate to near the initial value of the scenario to simulate a “popping of the clot” [22].

#### 2.2.2. Determination of Hemostasis Factors for Use with HATRC

The hemostasis factor was determined by analysis of swine hemorrhage data sets. To simulate the internal hemostatic effects on hemorrhage rate, the change in uncontrolled hemorrhage rates over time of 12 WB-infused swine were evaluated over a 25-min period, starting from the maximum hemorrhage rate observed. One data set was excluded due to being incomplete. Hemorrhage volume was recorded at a sampling time of five seconds, and real time hemorrhage rates were estimated by taking the volume delta from one minute prior for each data point. Each data set was then normalized by the maximum bleed rate observed within that set. Linear regression curves were produced beginning with the first three points, then iteratively adding additional points until the R-squared value fell below 0.95. This point was identified as a transition separating a short period of rapid clotting, hereon referred to as phase I hemostasis, and a much slower phase II hemostasis, which extended to the end of the 25-min period. Using the Phase I regression slopes, the data sets were divided into two groups: eight with shallow slopes, or slower hemostasis rates, and three with steeper slopes, or faster hemostasis rates. Each group was then averaged, and the regression analysis was repeated on the averaged sets. Four regression slopes were ultimately used, a slow and fast hemostasis pairing of phase I and phase II regions. These slopes were then applied as the hemostasis factor when calculating the hemorrhage rate over time.

A description of the hemorrhage scenarios evaluated and the justification for their design is detailed below.

### 2.3. Hemorrhage Testing Scenarios: Overview and Rationale

Each scenario is summarized in Table 1 as to the duration, starting MAP, hemorrhage rate, whether hemostasis was occurring, infusate type and other unique features of each scenario. Unless otherwise stated, each scenario was capped at 30 min and the target MAP was 65 mmHg. Prior to running the next scenario, HATRC was reset at new starting conditions to remove any potential spillover effect of controller performance on one scenario to the next. The rationale and descriptions for each scenario are described below, grouped by scenario category: compressible hemorrhage scenarios, non-compressible hemorrhage scenarios, infusate swapover scenarios and coagulopathy scenarios.

#### 2.3.1. Compressible Hemorrhage Scenarios

These scenarios were set up for mimicking a large compressible hemorrhage that was stopped, but not before causing profound hypotension. Briefly, Scenario 1 began resuscitation at a low starting MAP of 45 mmHg but with no active bleed. This mimicked a common real-world scenario seen with severe extremity trauma and subsequent tourniquet application, which would have stopped ongoing bleeding. The controller had 30 min to stabilize at a target MAP of 65 mmHg using WB as the infusate type, at which point a fast bleed rate was triggered for two minutes to mimic a tourniquet slippage event that was then corrected. The resuscitation continued for a total time of 60 min to allow time for re-stabilization. Scenario 2 began the same as the first scenario, except crystalloid was used for the infusion fluid type. This changed the volume responsiveness but, due to the design of PV_Crys_ accounting for fluid compartmentalization in the hydrostatic column, re-bleeds at high pressure were not physiologically relevant for crystalloid when using HATRC [16]. As a result, the second part of the scenario mimicking tourniquet slippage was not evaluated for crystalloid infusion.

#### 2.3.2. Non-Compressible Hemorrhage Scenarios

The next scenarios simulate bleed situations that were not manually stopped but were subjected to internal hemostatic mechanisms. Overall, these scenarios either started at the target MAP or a low MAP, to mimic the casualty being either manually resuscitated prior to the controller’s start or presenting with profound hypotension, respectively. In either case, both types of scenarios also mimicked a hemorrhage that was still ongoing. The initial hemorrhage rate was defined by a slow (Scenarios 3, 4, 5) or fast (Scenarios 6, 7, 8) rate equaling 70 or 140 mL/min at 65 mmHg, and the infusate was either WB (Scenarios 3, 4, 6, 7) or crystalloid (Scenarios 5, 8). Hemostasis was in effect throughout each scenario, which continued reducing the hemorrhage rate across the 30-min test window. As in Scenario 2, the high starting MAP paired with crystalloid was not physiologically relevant with HATRC as the pressure is volume unresponsive. As a result, these scenario types were not evaluated.

#### 2.3.3. Infusate Swapover Scenario

Scenario 9 evaluated resource-limited hemorrhage resuscitation. Specifically, it simulated a casualty presenting with a low initial MAP and a non-compressible, fast hemorrhage, in a situation where WB was limited to 2 units (~900 mL), so after this volume was reached, the infusion was swapped to crystalloid for the rest of the resuscitation. In order to simulate this infusate swapover, when the test scenario reached this point, HATRC paused resuscitation and hemorrhage, switched all solenoid valves from PV_WB_ to PV_Crys_, and filled PV_Crys_ until pressure was equal to PV_WB_ after the 900 mL infusion. After equalizing pressure, the test scenario resumed with crystalloid infusion for the remainder of the 30 min.

#### 2.3.4. Coagulopathy Scenarios

The final scenarios added a coagulopathy complexity where internal hemostatic mechanisms halt, or never start, so that any non-compressible bleed continued throughout the scenario. Scenario 10 used WB as the infusate and began at 45 mmHg MAP, with a slow hemorrhage and normal hemostasis. However, to stress the controller, after five minutes the hemostasis stopped, and the bleed rate began to intensify again up to a fast hemorrhage rate. For Scenario 11, crystalloid infusion was used and a fast bleed began at a starting MAP of 45 mmHg. In contrast to Scenario 10, this scenario experienced no hemostasis across its entire duration, so the rate remained at the maximum. This was to further stress the controller with more extreme trauma scenarios.

### 2.4. Control Systems Performance Evaluation

To evaluate controller performance, a series of performance metrics were calculated for each testing scenario and each controller configuration. Many of these metrics have been previously described but briefly, they are a combination of metrics from Varvel et al. [23], Mirinejad et al. [12], Marques et al. [11], Snider et al. [15] and IEC 60601-1-10 [24]. Each is summarized in Table 2 with descriptions of how they were calculated and specific references. We introduce a new variable called “area below target”, which is the sum of the negative component of the difference between the measured MAP and the target MAP accumulated throughout the test run. It accounts for both severity and duration of hypotension and is described in severity-adjusted minutes. This value is expected to somewhat represent overall oxygen debt, assuming no use of vasopressors and a close correlation between MAP and CO in this MAP range. All metrics were calculated using Python scripting to automate the analysis for each testing scenario.

Each controller was evaluated with all test scenarios, as detailed in Table 1, for three test runs. For the triplicate runs, subject variability was added to HATRC three ways. First, PV_WB_ and PV_Crys_ volume responsiveness was increased or decreased, creating three separate subjects for evaluation (high, normal, and low responsiveness for both PVs) [15]. Second, the coagulation and coagulopathy rates were adjusted between two possible speeds, as determined from the swine data sets as described above (Section 2.2.2). Third, the hemorrhage rate calculations have ±5% noise added, which changed the overall bleeds experienced throughout each iteration. In addition, triplicate runs for each scenario and controller configuration were summarized into test metrics. Metric data and subsequent analysis and graphical representation were performed using MATLAB and GraphPad Prism 9 (San Diego, CA, USA).

## 3. Results

### 3.1. Compressible Hemorrhage Scenarios

Scenarios 1 and 2 both mimicked a large hemorrhage that was able to be stopped by simulated compression or tourniquet usage. Both scenarios began with a low MAP (45 mmHg) but had no active bleed. The controllers had 30 min to stabilize the patient back to a target MAP of 65 mmHg. Scenario 1 used WB as the infusate type and incorporated a tourniquet slippage, while Scenario 2 used crystalloid and had no tourniquet slippage (Table 1).

#### 3.1.1. Scenario 1 Results

For Scenario 1, the performance differences were noticeable from the mean pressure and infusion flow rate versus time trends between the conservative and aggressive PID (Figure 2A–D). The aggressive configuration reached target pressure quicker, which was captured by rise time efficiency (Figure 2E, 3.17 min Cons. vs. 2.22 min Agg.) and effectiveness (Appendix A, 92.8% Cons. vs. 96.2% Agg.) performance metrics. While overall MAP recovery was improved, the aggressive tuning had a higher noise level, as indicated by MDAPE (Figure 2F, 0.58% Cons vs. 2.04% Agg) and wobble (Appendix A, 0.35% Cons vs. 0.56% Agg) being higher for the aggressive tuned PID. Further, the aggressive configuration was prone to target overshooting (Figure 2G, 1.07% Cons vs. 3.50% Agg), and MAP remained at an overshoot since no active bleed was present. This was further captured by the area above the target metric (Appendix A, 0.17 min Cons vs. 0.98 Agg). As a result of the overshoot, the aggressive tuning was diverging from the set point while the conservative tuning was still converging, as measured by the end-state divergence performance metric (Figure 2H, −0.41%/hr Cons vs. 0.51%/hr Agg). A summary of all test metrics for Scenario 1 is shown in Appendix A.

#### 3.1.2. Scenario 2 Results

For the next scenario, the same logic was repeated for crystalloid infusate, but no tourniquet slippage occurred after 30 min (Table 1). Overall, the performance trends were similar, except MAP recovery was slower given the lower volume responsiveness when infusing with crystalloid. The rise time efficiency (Appendix A, 3.6% Cons vs. 2.4% Agg) and effectiveness (Appendix A, 79.1% Cons vs. 88.6% Agg) highlighted the much quicker recovery for the aggressive tuned PID. In fact, the conservative tuned PID did not reach 65 mmHg in any replicates by the 30-min mark for this scenario (Appendix A), but the conservative PID was still converging on the target, while the overshoot by the aggressive PID resulted in divergence from the target, as highlighted by the end-state divergence (Appendix A, −4.4%/hr Cons vs. 2.5%/hr Agg). In contrast to Scenario 1, the aggressive PID had lower MDAPE compared to the conservative PID (Appendix A, 2.7% Cons vs. 1.2% Agg), likely due to the conservative PID struggling to reach the target. A summary of all test metrics for Scenario 2 is shown in Appendix A and MAP and flow rate vs. time plots are shown in Appendix A.

### 3.2. Non-Compressible, Internal Hemorrhage Scenarios

The next series of scenarios simulated bleed situations that were stopped via clotting instead of compression. Each of these scenarios began with either a stable MAP of 65 (Scenarios 3 and 6) or a low MAP of 45 (Scenarios 4, 5, 7, 8). These simulated patients then experienced a slow bleed rate maximum of 70 mL/min in Scenarios 3, 4 and 5, while the conditions called for a fast bleed rate maximum of 140 mL/min in Scenarios 6, 7 and 8. The infusate type was also adjusted between scenarios: WB was used in Scenarios 3, 4, 6 and 7 and crystalloid was used for Scenarios 5 and 8. In addition, all the scenarios in this group showed the PID controllers’ effect on changing hemorrhage rates as the internal hemostatic mechanisms were simulated throughout the time course (Table 1).

#### 3.2.1. Scenario 3 Results

Internal hemorrhage scenarios were varied by their starting MAP, hemorrhage rate and infusion type. Scenario 3 used 65 mmHg (simulating a casualty being connected to the system after initial manual resuscitation), slow hemorrhage rate and WB, respectively. The rapid responsiveness of both PID controllers stopped the MAP from drifting far from the target, with MAP never dropping below 60 mmHg (Figure 3A–D). While the hypovolemic burden was minimal, the aggressive tuning recovered more quickly, keeping the magnitude of area below the target small (Figure 3E, −0.61 min Cons vs. −0.23 min Agg). This was further highlighted by the MDAPE metric, as the error from target was higher with the conservative configuration (Figure 3F, 1.99% Cons vs. 0.85% Agg) due to its less decisive response to the decrease in MAP. The overall flow rates were similar with both controller configurations, but the aggressive PID signaled quicker flow rates on average and with higher flow rate variability (Figure 3G, 2.94% Cons vs. 22.6% Agg). The rapid flow rate changes did impact the overall volume efficiency (Figure 3H, 0.93 Cons vs. 1.04 Agg), with the conservative configuration more efficiently stabilizing this scenario. A summary of all test metrics for Scenario 3 is shown in Appendix A.

#### 3.2.2. Scenario 4 Results

Scenario 4 started at 45 mmHg MAP with a slow hemorrhage rate and WB infusate. Only the aggressive controller overshot the target (Appendix A, 0.92% Cons vs. 2.2% Agg) in this scenario and had a higher average infusion rate (Appendix A, 39.7 mL/min Cons vs. 42.6 mL/min Agg) and infusion rate variability (Appendix A, 7.2% Cons vs. 29.6% Agg). Similar to the other scenarios, the aggressive configuration reached the target quicker (Appendix A, rise time efficiency, 3.6 min Cons vs. 2.3 min Agg) and reduced the hypotensive burden (Appendix A, area below target, −1.36 min Cons vs. −0.69 min Agg). However, due to target overshoot, the aggressive configuration was diverging from the target as the set point was reached (Appendix A), while the conservative configuration was closer to steady state (Appendix A, end-state divergence, 0.65%/hr Cons vs. 7.6%/hr Agg). A summary of all test metrics for Scenario 4 is shown in Appendix A and MAP and flow rate vs. time plots are shown in Appendix A.

#### 3.2.3. Scenario 5 Results

Scenario 5 began at 45 mmHg MAP with a slow hemorrhage rate and crystalloid infusate. With this change to crystalloid, one apparent result was with volume efficiency having a larger difference between the two PID configurations (Appendix A, 3.61 Cons vs. 4.10 Agg). Due to the lower volume responsiveness, the slower flow rates selected by the conservative configuration (Appendix A, mean infusion, 55.0 mL/min Cons vs. 66.9 mL/min Agg) resulted in a stark area below the target (Appendix A, −1.89 min Cons vs. −1.05 min Agg) and an overall rise time efficiency difference (Appendix A, 3.83 min Cons vs. 2.67 min Agg). A summary of all test metrics for Scenario 5 is shown in Appendix A and MAP and flow rate vs. time plots are shown in Appendix A.

#### 3.2.4. Scenario 6 Results

Scenario 6 used a fast hemorrhage rate and WB infusate and started at an initial MAP of 65 mmHg (Table 1). This scenario was identical to Scenario 3 but with a more severe hemorrhage rate. While the hemorrhage rate was more severe, the overall data trends were similar. MDAPE (Appendix A, 3.7% Cons vs. 1.6% Agg), area below the target (Appendix A, −1.14 min Cons vs. −0.52 min Agg), infusion rate variability (Appendix A, 3.7% Cons vs. 26.0% Agg) and volume efficiency (Appendix A, 0.96 Cons vs. 1.00 Agg) had the same trends compared to Scenario 3. A summary of all test metrics for Scenario 6 is shown in Appendix A and MAP and flow rate vs. time plots are shown in Appendix A.

#### 3.2.5. Scenario 7 Results

Mirroring Scenario 4, this scenario used a lower starting MAP of 45 mmHg and WB infusate but a fast hemorrhage rate. Overall infusion rates output by the controllers were quicker to combat the increased hemorrhage rate, but trends between the controller configurations were similar, with the aggressive PID having a higher infusion rate (Appendix A, 52.4 mL/min Cons vs. 57.0 mL/min Agg) and infusion rate variability (Appendix A, 6.8% Cons vs. 32.5% Agg). Rise time efficiency (Appendix A, 4.22 min Cons vs. 2.58 min Cons) and hypovolemic debt (Appendix A, Area below target, −1.80 min Cons vs. −0.91 min Agg) were similar to the less severe hemorrhage rate scenario, highlighting the PID’s capability to adapt therapeutic rates to different scenarios. A summary of all test metrics for Scenario 7 is shown in Appendix A and MAP and flow rate vs. time plots are shown in Appendix A.

#### 3.2.6. Scenario 8 Results

Scenario 8 was a repeat of Scenario 5, with a 45 mmHg starting MAP and crystalloid infusate, with one exception—a fast hemorrhage rate. Once again, the severe hemorrhage rate had minimal impact when comparing the controller’s performance as the PID methodology accounted for the increased rate (Appendix A). Aggressive tuning continued to out-perform conservative in terms of minimizing hypovolemic debt (Appendix A, Area below target, −2.27 min Cons vs. −1.23 min Agg) and rise time efficiency (Appendix A, 4.36 min Cons vs. 3.00 min Agg), but it still slightly overshot the target (Appendix A, 0.04% Cons vs. 0.71% Agg) and consistently had a high infusion rate variability (Appendix A, 6.5% Cons vs. 35.0% Agg), unlike the conservative configuration. A summary of all test metrics for Scenario 8 is shown in Appendix A and MAP and flow rate vs. time plots are shown in Appendix A.

### 3.3. Swapover Scenario 9 Results

An added feature of the HATRC system is the capability of swapping between infusates. This scenario was evaluated by starting at 45 mmHg, using WB infusate and a fast bleed rate. After 900 mL was infused, the infusion was swapped from WB to crystalloid to mimic resource-limited hemorrhage resuscitation. From the MAP vs. time plot, the swapover between systems and subsequent fluid responsiveness between infusate types can be seen (Figure 4A–D). Overall effectiveness of the resuscitation (Appendix A, 74.4% Cons vs. 88.7% Agg) and rise time efficiency (Figure 4E, 3.53 min Cons vs. 2.36 min Agg) were much higher for the aggressive configuration. This was more pronounced for this scenario as the closer to the target the controller reached while using the WB infusate, more of the resuscitation was performed in a more volume responsive state. The difference in hypovolemic debt, or area below the target, (Figure 4F, −2.03 min Cons vs. −1.17 min Agg) reflects this. The overall MDAPE (Figure 4G, 4.86% Cons vs. 2.09% Agg) was reduced with the aggressive configuration and it was much closer to steady state than the conservative configuration, as measured by end-state divergence (Figure 4H, −11.0%/hr Cons vs. 1.54%/hr Agg). A summary of all test metrics for Scenario 9 is shown in Appendix A.

### 3.4. Coagulopathy Scenarios

#### 3.4.1. Scenario 10 Results

The next scenario added further complexities to the testing in the form of coagulopathy. Scenario 10 used a WB infusate and initial hemostasis simulation that ceased after 5 min, followed by a hemorrhage rate increase until it reached the maximum for a fast rate of 140 mL/min. The MAP vs. time plots show how the two controllers responded to this additional load, with neither being able to reach the precise target MAP within 30 min (Figure 5A–D). The aggressive controller got closer to the target, as highlighted by effectiveness (Figure 5E, 14.2% Cons vs. 90.1% Agg) and MDAPE (Figure 5F, 10.75% Cons vs. 5.16% Agg) The hypovolemic burden was, as expected, higher with the conservative tuning (Figure 5G, −3.29 min Cons vs. −1.77 min Agg). However, for this scenario it was evident that the conservative PID configuration was further from steady state and more slowly converging to the target MAP, as was reflected by the end-state divergence metric (Figure 5H, −3.38%/hr Cons vs. −5.92%/hr Agg). A summary of all test metrics for Scenario 10 is shown in Appendix A.

#### 3.4.2. Scenario 11 Results

The final scenario looked at a similar coagulopathic state but with crystalloid as the infusate and no hemostasis throughout the entire test scenario. Both controllers were unable to reach the target again but, interestingly, they reached the same approximate MAP value after 30 min (Appendix A). However, the overall effectiveness for the conservative configuration was 0%, compared to 73.3% for the aggressive configuration (Appendix A). Similar to the other coagulopathy scenario, both controllers trended toward convergence, but more time was likely needed to reach the target MAP (Appendix A, end-state divergence, −4.20%/hr Cons vs. −1.93%/hr Agg). Due to neither controller reaching the set point, the hypovolemic burden was large for both, but larger for the conservative PID (Appendix A, area below target, −4.21 min Cons vs. −2.39 min Agg). A summary of all test metrics for Scenario 11 is shown in Appendix A and MAP and flow rate vs. time plots are shown in Appendix A.

### 3.5. Summary Data

Lastly, we evaluated performance across all the testing scenarios to quantify the overall takeaway trends. It was evident that MDAPE was reduced in most scenarios for the aggressive PID compared to conservative (Figure 6A, 4.78% Cons vs. 2.27% Agg). A similar consistent difference was evident for the area below the target (Figure 6B, −1.94 min Cons vs. −1.05 min Agg), end-state divergence (Figure 6C, −4.75%/hr vs. 2.43%/hr Agg) and rise time efficiency (Figure 6D, 3.26 min Cons vs. 2.35 min Agg). However, the conservative configuration outperformed the aggressive tuning in terms of target overshoot (Figure 6E, 0.41% Cons vs. 1.25% Agg) and infusion rate variability (Figure 6F, 6.0% Cons vs. 30.79% Agg). Selecting a proper controller design will be influenced by a number of performance metrics and will ultimately depend on metric prioritization, based on how the desired performance is defined.

## 4. Discussion

Fluid resuscitation for hemorrhagic shock is an effective treatment if properly administered, but that is not always available. Goal directed damage control resuscitation requires constant monitoring and adjustments to the infusion to ensure the balance between organ perfusion and re-bleeding prevention. This becomes even more challenging in resource-limited remote or military medicine scenarios where medical personnel are stretched thin and lack expertise. Different casualties with different injuries present with different fluid responsiveness patterns, requiring an adaptive, personalized approach with tight control of their infusion rate. For these reasons, closed-loop fluid administration systems can have a major impact on improving trauma care, by automating the required personalized medical treatment needed to properly stabilize a hemorrhagic shock patient. Here, we extensively detail how a PID controller can be used for MAP-based goal-directed fluid resuscitation through a number of tailored testing scenarios.

The battery of scenarios was critical for properly identifying strengths and weaknesses in controller designs. These scenarios were selected based on the capabilities of the test platform and to mimic realistic scenarios, as well as more extreme situations meant to challenge the controllers’ performance boundaries. All of this was possible using the HATRC test platform, which is a physical platform with sensors and actual fluid infusion, as opposed to computer simulations. This allowed better accounting of pump and sensor noise into the system without removing physiological mimicking volume responsiveness during resuscitation or even subject variability. The eleven scenarios used in this study paired changes to the hemorrhage rate, hemostasis rate and infusion type, as well as coagulopathy or infusate changes mid-resuscitation, to create each scenario. The performance and decisions made the aggressive PID controller in Scenario 1 significantly overshoot the target while with Scenario 11, the controller could not reach the precise target pressure. Through these different scenarios, the personalization ability of PID controllers was evident and the variability in results highlighted the importance of robust testing, which may not be cost-effective or feasible in live animal studies.

Through these testing scenarios, performance trends between the two controller configurations were evident. The aggressive controller was capable of overshooting the target, especially in low or non-hemorrhage situations. However, the magnitude of the overshoot, while clearly evident, might not be clinically significant. On a similar note, the severity of the hemorrhage had minimal effect on the PID controllers’ performance. In particular, we evaluated more extreme hemorrhage scenarios as those would likely be challenges for a controller, but that was not the case for the PID controllers. This was initially a surprising result as, intuitively, more severe hemorrhages are innately harder to handle, but the proportional aspect of the controller appropriately scaled flow rate, and the resulting data differences between the fast and slow hemorrhage rates were minimal. This is unlikely to be the case for other controller types such as decision table-based designs. One expected trend for the study was that the conservative controller was slower to reach the target, associated with a greater hypotensive burden, and less likely to overshoot as a result, but in later scenarios this configuration was struggling to reach the target set point, resulting in less than 50% effectiveness in the coagulopathy scenarios. With each controller having its own challenges and positive design features, it leaves the question as to how to identify which controller is best for hemorrhagic shock resuscitation?

We believe the answer to that question lies with the use case for the system. Different patient conditions and care scenarios will require different controller configurations. Further study is required to determine the optimal resuscitation profile from a physiologic standpoint. Other trauma scenarios may require different controller configurations and PID controllers can be tuned for those personalized specifications. However, for this work, we focused on these aggressive and conservative configurations. Defining these two use cases was subjective though, and we believe that this can be improved by standardizing the summary metrics that aggregate the criteria that are most essential for each use case. For aggressive tuning, three of the four metrics seem most critical to optimize against: area under the target, rise time efficiency and effectiveness or MDPE. The goal for aggressive resuscitation would be to reach the target as quickly as possible and an aggregate of each of these metrics will reflect that goal. For conservative tuning, two metrics seem most critical: target overshoot and volume efficiency. The goal for tuning in these scenarios is stability, limiting resources and avoiding overshooting the target, which these metrics help to identify. Other metrics can be included, or weights added to more critical features, but aggregate metrics will assist with interpreting nuanced results between metrics when comparing controller types or scenarios.

While the scenarios helped with identifying strengths and weaknesses for each controller configurations, there are some weaknesses with their current design. First, the scenarios were developed from retrospective swine data. While the data sets were extensive, they were limited in terms of the volume infused and differences in hemorrhage rates. This required extrapolation of rates and potential issues with overfitting the results to these datasets. Second, while subject variability was introduced to the coagulation rates, hemorrhage rate and volume responsiveness, variability can be expanded to make the test platform more robust. Third, the timeframe for each scenario was limited to 30 min, with the exception of Scenario 1 that had two distinct phases. Longer times may have allowed more controller scenarios to reach their target or revealed additional strengths and weaknesses with their performance. However, a set time frame was selected to streamline the testing while still allowing head-to-head evaluation. Lastly, only two controller configurations were considered at this time. More PID configurations could help tease out performance details or better optimize the aggressive and conservative use cases, but that was outside the scope of this current work.

Next steps for this work will take three approaches. First, a comparison of various controller types, such as decision table, fuzzy logic, or adaptive designs, will be conducted to better highlight which controller logics are ideal for certain scenarios. Second, tuned controllers will be used in live animal studies as an external validation step for the test platform and to refine the model with more robust animal data. Third, vasopressor controllers will be evaluated with the test platform, as we have previously shown that the HATRC test platform can be used for simulating vasopressor action in the testbed.

## 5. Conclusions

In summary, closed-loop control systems have the potential to personalize hemorrhagic shock fluid resuscitation to each casualty, if properly designed. Here, we show how the HATRC testbed can be utilized for evaluating PID controllers across a wide range of hemorrhage scenarios. The two controllers evaluated were tuned for either more aggressively reaching the target at the expense of potential overshooting, or tuned more conservatively, gently resuscitating at the expense of the pressure remaining low for longer. We recommend that aggregate metrics for these two situations be identified and utilized to more optimally design controllers for these unique situations. In conclusion, PID controllers are capable of responding uniquely to complex hemorrhage situations and can be tuned to meet the required use case. The HATRC test platform can assist with development and optimization of closed-loop controllers prior to animal studies, to accelerate the transition pipeline for fluid administration controllers and make automating this challenging medical process into more of a reality.

## Figures and Tables

**Figure 1 jpm-12-00979-f001:**
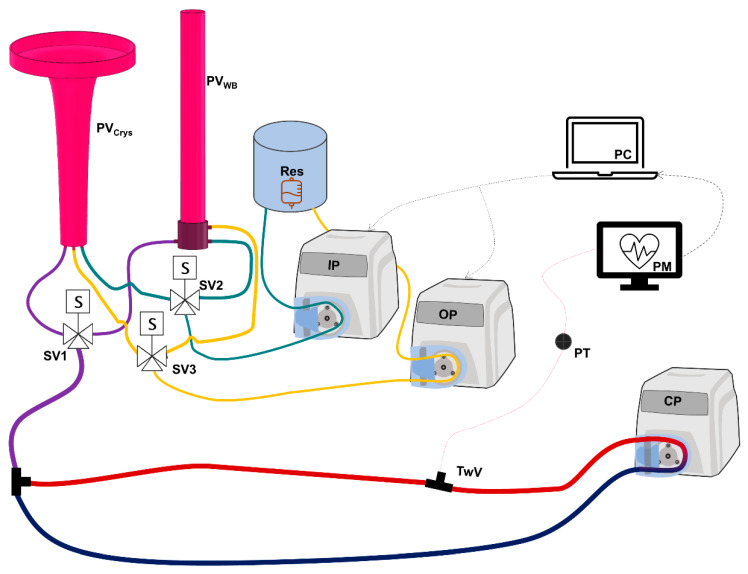
**Diagram of HATRC Flow Loop**. The arterial and venous side of the main circulatory flow loop are colored red and blue, respectively. Circulatory flow is supplied by a peristaltic pump (CP). Then, arterial pressure is measured by pressure transducer (PT) through fluidic connection at a three-way valve (TwV). Arterial waveform data are displayed with a patient monitor (PM) and recorded in real-time via computer (PC) running the hemorrhage scenarios and PID controllers. Static pressure is supplied to the loop via purple line from either whole blood (PV_WB_) or crystalloid (PV_Crys_) PhysioVessels based on position of solenoid valve (SV1). Infusion (IP) and outflow (OP) peristaltic pumps are supplied with water from an external reservoir (Res) and are connected via teal and yellow lines, respectively, to add or remove volume from the appropriate PhysioVessel based on position of SV2 and SV3. Both peristaltic pumps are fully controlled by the algorithm running on the PC.

**Figure 2 jpm-12-00979-f002:**
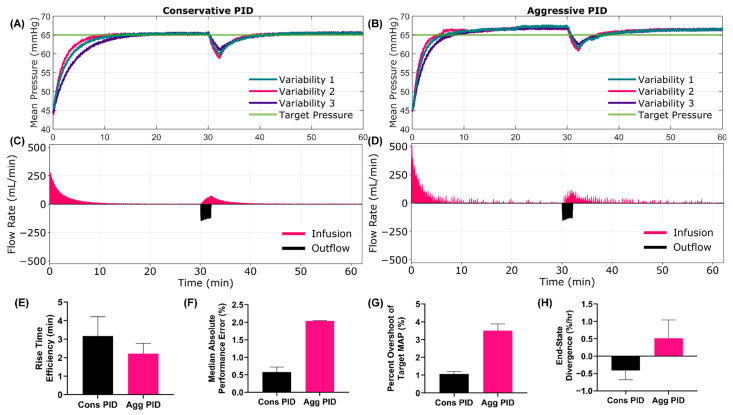
**Scenario 1 Testing Results**. Scenario 1 began at a low starting MAP with no active bleed, which mimicked tourniquet usage to stop an extremity bleed. At 30 min, the tourniquet slipped which resulted in a high hemorrhage rate that stopped after 120 s. WB infusate was used for this scenario. Results for the conservative (**A**,**C**) and aggressive (**B**,**D**) tuned PID are shown for (**A**,**B**) MAP and (**C**,**D**) infusion and outflow rates versus time as the controller tried to reach a target MAP of 65 mmHg. For MAP plots, each of three subject variability runs are plotted individually, while a single representative run is shown for the inflow/outflow plots. Average controller performance metrics for (**E**) rise time efficiency, (**F**) MDAPE, (**G**) target overshoot and (**H**) end-state divergence are shown for both controller tunings for this scenario. Error bars denote standard deviation (*n* = 3).

**Figure 3 jpm-12-00979-f003:**
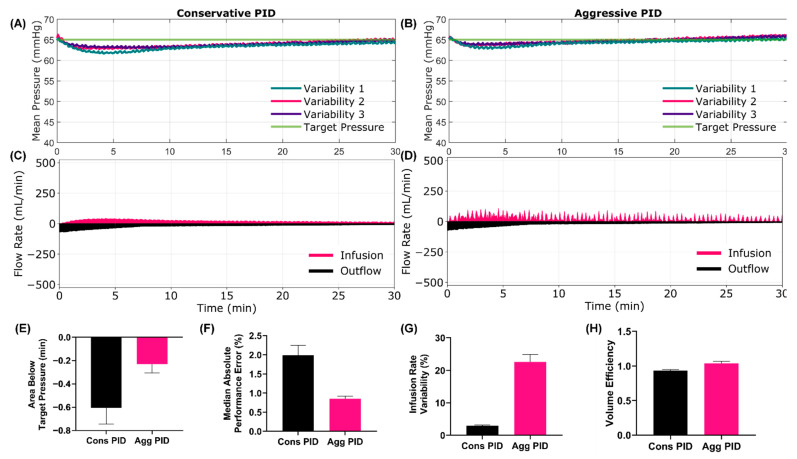
**Scenario 3 Testing Results**. Scenario 3 began at a stable 65 mmHg starting MAP with a slow active bleed that clotted over the 30-min testing scenario. WB infusate was used for this scenario. Results for the conservative (**A**,**C**) and aggressive (**B**,**D**) tuned PID are shown for (**A**,**B**) MAP and (**C**,**D**) infusion and outflow rates versus time, as the controller tried to maintain a target MAP of 65 mmHg. For MAP plots, each of three subject variability runs are plotted individually, while a single representative run is shown for the inflow/outflow plots. Average controller performance metrics for (**E**) area below target pressure, (**F**) MDAPE, (**G**) infusion rate variability and (**H**) volume efficiency are shown for both controller tunings for this scenario. Error bars denote standard deviation (*n* = 3).

**Figure 4 jpm-12-00979-f004:**
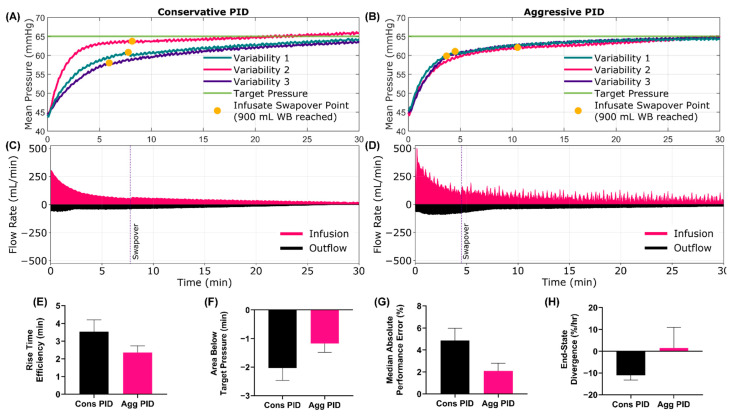
**Scenario 9 Testing Results**. Scenario 9 began at a low starting MAP with an active fast bleed, which clotted over time. This scenario mimicked resource-limited situations where fluid infusion switched from WB after 900 mL (~2 units) to crystalloid for the remainder of time. Results for the conservative (**A**,**C**) and aggressive (**B**,**D**) tuned PIDs are shown for (**A**,**B**) MAP and (**C**,**D**) infusion and outflow rates versus time as the controller tried to reach a target MAP of 65 mmHg. For MAP plots, each of three subject variability runs are plotted individually, while a single representative run is shown for the inflow/outflow plots. Average controller performance metrics for (**E**) rise time efficiency, (**F**) area below target pressure, (**G**) MDAPE and (**H**) end-state divergence are shown for both controller tunings for this scenario. Error bars denote standard deviation (*n* = 3).

**Figure 5 jpm-12-00979-f005:**
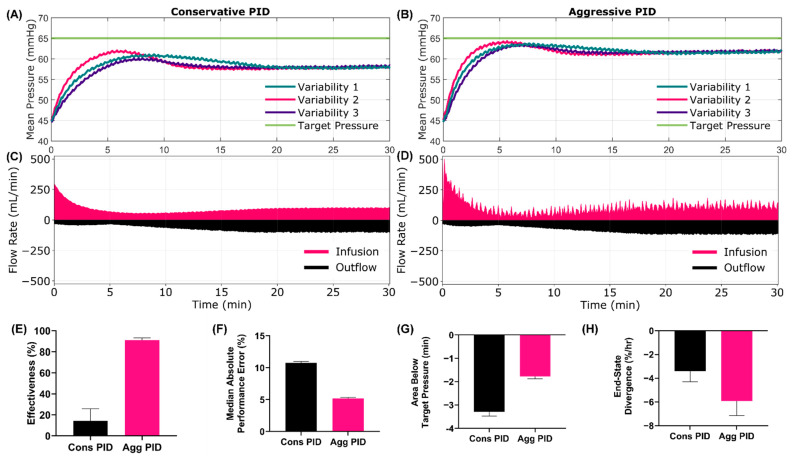
**Scenario 10 Testing Results.** Scenario 10 began at a low starting MAP with an active fast bleed, which clotted for 5 min when coagulopathy caused the hemorrhage rate to increase up to a maximum of 140 mL/min, where it stayed for the remainder of the testing scenario. WB infusate was used for this scenario. Results for the conservative (**A**,**C**) and aggressive (**B**,**D**) tuned PID are shown for (**A**,**B**) MAP and (**C**,**D**) infusion and outflow rate as the controller tried to reach a target MAP of 65 mmHg. For MAP plots, each of three subject variability runs are plotted individually while a single representative run is shown for the inflow/outflow plots. Average controller performance metrics for (**E**) effectiveness, (**F**) MDAPE, (**G**) area below target pressure and (**H**) end-state divergence are shown for both controller tunings for this scenario. Error bars denote standard deviation (*n* = 3).

**Figure 6 jpm-12-00979-f006:**
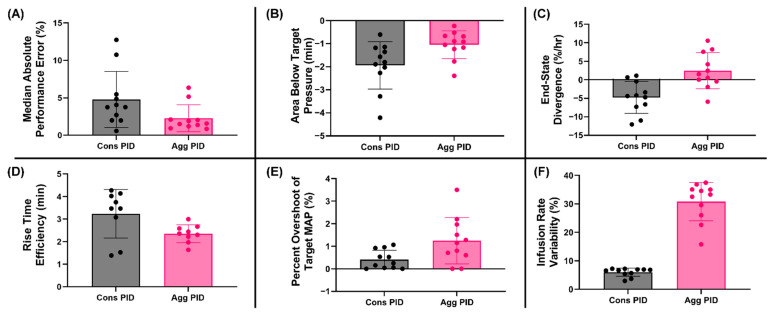
**Summary of HATRC testing for PID tuning.** Summary plots across the 11 testing scenarios for (**A**) MDAPE, (**B**) area below the target pressure, (**C**) end-state divergence, (**D**) rise time efficiency, (**E**) target overshoot and (**F**) infusion rate variability. Data points represent mean results for each scenario (*n* = 11). Only 9 scenarios were included for rise time efficiency as this metric cannot be calculated for scenarios 3 and 6 where MAP started at 65mmHg.

**Table 1 jpm-12-00979-t001:** Summary of hemorrhage scenarios used for evaluating PID controllers in HATRC.

	**Duration (min)**	**Starting MAP**	**Initial Hemorrhage Rate**	**Hemostasis?**	**Coagulopathy?**	**Infusate Type**	**Infusate Swapover?**
Compressible Hemorrhage Scenarios							
Scenario 1a	30	45 mmHg	None	NA	No	WB	No
Scenario 1b	30	End MAP 1a	Fast, 120 s Followed by None	NA	No	WB	No
Scenario 2	30	45 mmHg	None	NA	No	Crystalloid	No
Non-Compressible Hemorrhage Scenarios							
Scenario 3	30	65 mmHg	Slow	Yes	No	WB	No
Scenario 4	30	45 mmHg	Slow	Yes	No	WB	No
Scenario 5	30	45 mmHg	Slow	Yes	No	Crystalloid	No
Scenario 6	30	65 mmHg	Fast	Yes	No	WB	No
Scenario 7	30	45 mmHg	Fast	Yes	No	WB	No
Scenario 8	30	45 mmHg	Fast	Yes	No	Crystalloid	No
Infusate Swapover Scenarios							
Scenario 9	30	45 mmHg	Fast	Yes	No	WB → Crystalloid	Yes, at 900 mL WB
Coagulopathy Scenarios							
Scenario 10a	5	45 mmHg	Slow	Yes	No	WB	No
Scenario 10b	25	End MAP of 10a	Fast	No	Yes, increasing hemorrhage rate until Fast	WB	No
Scenario 11	30	45 mmHg	Fast	No	Yes, hemorrhage rate held at Fast	Crystalloid	No

**Table 2 jpm-12-00979-t002:** Summary of controller performance metrics evaluated in each testing scenario.

Testing Metric	Description	Ref.
Median performance error (MDPE)	Median performance error (% error from target pressure) across the test scenario	[12,23]
Median absolute performance error (MDAPE)	Median of the absolute value of the performance errors across the test scenario	[23]
Steady-state overshoot	Maximum MAP reached after reaching steady-state relative to the steady-state MAP	[12,23]
Steady-state undershoot	Minimum MAP reached after reaching steady-state relative to the steady-state MAP	
Target overshoot	Maximum MAP reached relative to the target MAP	[15]
Effectiveness	Percentage of time MAP was within +/− 5 mmHg of the target pressure	[11]
Wobble	Median of the absolute value for the difference between performance error and MDPE across the test scenario	[12,23]
End-state divergence	Slope of MDAPE vs. time for the final 10% of each testing scenario. (Note: The calculation was limited to this time range to evaluate the final controller trends at the end of each scenario)	[12,15]
Rise time efficiency	Amount of time required for MAP to reach 90% of steady-state MAP	[11,24]
Volume efficiency	Ratio between volume infused and volume lost	[11]
Median infusion rate	Median of the infusion rates across the test scenario	[15]
Mean infusion rate	Arithmetic mean of the infusion rates across the test scenario	[15]
Area above target pressure	Area above the target pressure while below the MAP vs. time curve for each test scenario, relative to target MAP	[15]
Area below target pressure	Area below the target pressure while above the MAP vs. time curve for each test scenario, relative to target MAP, representing a “hypotensive burden”.	[15]
Area to rise time	Area below target pressure until rise time was reached (90% of the target pressure), relative to target MAP.	[15]
Infusion rate variability	Average of the 2-min binned standard deviations of the infusion rate relative to the mean across each scenario	

## Data Availability

The datasets generated during and/or analyzed during the current study are available from the corresponding author upon reasonable request.

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
