# Peer review of "Evaluation of a Proportional–Integral–Derivative Controller for Hemorrhage Resuscitation Using a Hardware-in-Loop Test Platform"

_jpm, 2022, doi:10.3390/jpm12060979_

Round 1

Reviewer 1 Report

The authors developped a hardware-in-loop test platform for the simulation of different bleeding scenarios. Based on this platform different  integral derivative controllers for hemorrhage resuscitation were tested. Automated hemorrhage resuscitation is of interest. However, I have some concerns and questions about the used platform. It seems that the platform is too simple to be an adequate model for a complex physiologic system.

1) How does the platform simulate the change of systemic vascular resistance?

2) How does the platform simulate the frank-starling-mechanism?

3) Whole blood and crystalloids remain in the vascular system for a different length of time. An adequate resuscitation is more dependent from the used type of fluid than solely on the given volume. How does the system simulate different types of fluid/ vascular pemeability?

Please provide more information about the used test platform, as the relevant citation has not yet passed a peer review process:

Snider, E.J.; Berard, D.; Vega, S.J.; Hernandez-Torres, S.; Avital, G.; Boice, E.N. Hardware in Loop Automated Testbed for Evaluating Hemorrhagic Shock Fluid Resuscitation Controllers. Scientific Reports Submitted.

Author Response

Dear Ms. Moana Li,

We would like to thank you and the Reviewers for the thorough review of our manuscript and for giving us the opportunity to respond to the comments. We have reproduced the Reviewer comments below in italics, followed by our responses. Changes to the manuscript are tracked for the Reviewers’ convenience and are reproduced in our responses below when possible. We hope that our responses address all the Reviewers’ concerns and that our study is now suitable for publication.

Sincerely,       

Eric J. Snider

Reviewer A

  • How does the platform simulate the change of systemic vascular resistance?
  • How does the platform simulate the frank-starling-mechanism?
  • Whole blood and crystalloids remain in the vascular system for a different length of time. An adequate resuscitation is more dependent from the used type of fluid than solely on the given volume. How does the system simulate different types of fluid/ vascular permeability?

The answer to all of these questions lies in the initial design of the HATRC system. It is designed to link mean arterial pressure (MAP) response to infusion volume. In realistic physiology, these two are linked through several variables, including starting volume status (position on the Frank-Starling curve), cardiac contractility (shape of the Frank-Starling curve), total peripheral resistance, fluid type and vascular permeability, venous tone, etc. One approach is creating a system that simulates all the above, which has been done by different groups as an in-silico model, examining only the software component of the system. Such complexity with variable responses of each of those parameters is impractical to create as a hardware-in-loop platform. Therefore, we have employed the second approach – the empiric one. We have added reference to this empirical approach in the methods section.

We have used exploratory data from a hemorrhage-resuscitation live swine experiment conducted in our institute and extracted the MAP response to fluid infusion (crystalloids and whole blood separately) into mathematical functions. Not surprisingly, the response to whole blood infusion in these MAP ranges (simulating class III-IV shock) was approximately linear, while the response to crystalloids demonstrated a second-order relationship, most likely due to fluid re-distribution. These equations were translated into the shape “PhysioVessels” – containers simulating venous capacitance, with a contour representing the curve-shape of the above-mentioned equations, leading to a cylindrical shape for the “whole blood” PhysioVessel, and a convex funnel shape for the crystalloids PhysioVessel. Further description of these methods can be found in Ref. 16. Berard, D.; Vega, S.J.; Torres, S.I.H.; Polykratis, I.A.; Salinas, J.; Ross, E.; Avital, G.; Boice, E.N.; Snider, E.J. Development of the PhysioVessel: A Customizable Platform for Simulating Physiological Fluid Resuscitation. Biomed. Phys. Eng. Express 2022, 8, 035017,  doi:10.1088/2057-1976/ac6196.

To summarize, under the assumption of normal cardiac function, in the range of class III-IV hemorrhagic shock and MAP goals corresponding with damage control resuscitation, the PhysioVessels are designed to mimic the empiric MAP response of healthy pigs to either whole blood or crystalloid resuscitation. The choice for this design is derived from our mission to automize and optimize volume resuscitation for wounded soldiers, based on combat casualty care principles, which allows us to make these assumptions. Any extrapolation to different populations or different ranges on the Frank-Starling curve would require addition data or a change of approach.

  • Please provide more information about the used test platform, as the relevant citation has not yet passed a peer review process:

    Snider, E.J.; Berard, D.; Vega, S.J.; Hernandez-Torres, S.; Avital, G.; Boice, E.N. Hardware in Loop Automated Testbed for Evaluating Hemorrhagic Shock Fluid Resuscitation Controllers. Scientific Reports Submitted.

We have included the draft of this submitted manuscript to further expand on those details. But, the included Ref. 16 introduces the PhysioVessel and describes the governing equations, mechanical principles, and empirical data that informed its design. The authors felt that due to the dense nature of the current work, it would be best to forgo a more detailed description and summarized this component in lines 107-113. Lines 113-118 have been added to clarify the justification of using this model in the present application. Figure 1 has been added to the main document to present the complete HATRC system in diagram form.

Reviewer 2 Report

This study evaluated a closed loop controller for its ability to administer fluids and stabilize the HATRC platform at a target pressure across eleven testing scenarios.

The method is extremely complex and hard to understand readers of this journal, because we are not familiar with this HATRC test platform. I understand this platform is very important to test their novel closed loop control systems. However, it has not been described in detail. I could not find its reference article (#15 is still under submission to Sci Rep, and has not been published nor has it been even accepted). In addition, the authors should simplify these scenarios and their results.

Author Response

Dear Ms. Moana Li,

We would like to thank you and the Reviewers for the thorough review of our manuscript and for giving us the opportunity to respond to the comments. We have reproduced the Reviewer comments below in italics, followed by our responses. Changes to the manuscript are tracked for the Reviewers’ convenience and are reproduced in our responses below when possible. We hope that our responses address all the Reviewers’ concerns and that our study is now suitable for publication.

Sincerely,       

Eric J. Snider

Reviewer B

  • The method is extremely complex and hard to understand readers of this journal, because we are not familiar with this HATRC test platform. I understand this platform is very important to test their novel closed loop control systems. However, it has not been described in detail.

I could not find its reference article (#15 is still under submission to Sci Rep, and has not been published nor has it been even accepted).

While the HATRC test platform manuscript is still under review, a principal component of the HATRC system is the PhysioVessel, the development of which was published in Ref 16. We have modified the methods to build on the original PhysioVessel flow loop setup instead of HATRC. An updated diagram of the entire system has been added in Figure 1 in the main text to supplement the current absence of Ref. 15. Elaboration on the design and justifications for its use in the current application was added in lines 104-123.

  • In addition, the authors should simplify these scenarios and their results.

The scope of the manuscript was to highlight how the test platform allowed for a range of controller evaluations and how that can help tease out differences in controller performance. Because of that, the scenarios vary in complexity, from simple non-ongoing hemorrhage situations to more complex coagulopathy mimicking scenarios. The scenarios cannot be simplified further if we want to still be able to capture key physiological effects in each scenario. We have presented the scenario details in a number of different ways for easier comprehension - a summary in Table 1 of all the scenarios, text in the methods section, and results with accompanying figures. As for the results, we agree the data is dense for this research effort as there are many metrics, controllers and scenarios. The arduous metric tables are kept in the supplement as they were too complicated to keep track of in the main text and instead only key metrics and results are highlighted in the text and graphical figures for key scenarios. These were done to simplify the results as much as possible.

Round 2

Reviewer 1 Report

Thank you for your reply, my questions have been comprehensively answered. However, I have a further question. Your controller is using MAP as the primary resuscitation goal. In hemorrhage, it is also essential to ensure adequate hemoglobin levels without unnecessary over-transfusion. Up to now, the hemoglobin level has not been taken into account in the model. Why do you not consider this in your simulation as it is a crucial parameter during hemorrhagic shock resuscitation? 

Author Response

Dear Ms. Moana Li,

We would like to thank you again for the thorough review of our manuscript and for the additional comment. We have reproduced the Reviewer comment below in italics, followed by our response. We hope that our response addresses all the Reviewers’ concerns and that our study is now suitable for publication.

Sincerely,       

Eric J. Snider

Reviewer A

Thank you for your reply, my questions have been comprehensively answered. However, I have a further question. Your controller is using MAP as the primary resuscitation goal. In hemorrhage, it is also essential to ensure adequate hemoglobin levels without unnecessary over-transfusion. Up to now, the hemoglobin level has not been taken into account in the model. Why do you not consider this in your simulation as it is a crucial parameter during hemorrhagic shock resuscitation?

Thank you for your rapid response and thoughtful consideration of alternative uses for the HATRC platform. In the specific context of combat casualty care and remote damage control resuscitation, especially within the time frame and goals we used, hemoglobin concentration is usually not used as a guide for transfusion under the assumption that it doesn’t represent hemoglobin quantity, as what’s lost is whole blood and whole blood should be used as the resuscitation fluid. This is reflected in the joint trauma system’s guidelines. Once hemodynamic goals are achieved, then we start considering packed red blood cells with the goal of restoring hemoglobin concentration.

Additionally, we have explored modifications to the flow loop setup to incorporate additional sensors, such as the Massimo Rainbow sensor, and indocyanine green dye in the tap water to mimic hemoglobin concentration and dilution. The added benefit was not deemed critical to the performance metrics we wished to capture in this manuscript. It is, however, being considered for future iterations of flow loop design in which we will attempt to cover these more advances stages of care, and the issue of hemoglobin concentration will definitely require a solution such as the one you suggest.

Reviewer 2 Report

I have no more comments. Thanks.

Author Response

Dear Ms. Moana Li,

We would like to thank you again for the thorough review of our manuscript and rapid reponse.  We hope that our study is now suitable for publication.

Sincerely,       

Eric J. Snider

Round 3

Reviewer 1 Report

Thank you for your reply. I recommend to accept the manuscript for publication. Congratulations.